# A Vicious NGF-p75^NTR^ Positive Feedback Loop Exacerbates the Toxic Effects of Oxidative Damage in the Human Retinal Epithelial Cell Line ARPE-19

**DOI:** 10.3390/ijms242216237

**Published:** 2023-11-12

**Authors:** Giuseppe Tringali, Michela Pizzoferrato, Lucia Lisi, Silvia Marinelli, Lucia Buccarello, Benedetto Falsini, Antonino Cattaneo, Pierluigi Navarra

**Affiliations:** 1Section of Pharmacology, Department of Healthcare Surveillance and Bioethics, Catholic University Medical School, Fondazione Policlinico Universitario A. Gemelli IRCCS, 00168 Rome, Italymichela.pizzoferrato@gmail.com (M.P.);; 2European Brain Research Institute-Fondazione Rita Levi Montalcini, 00161 Rome, Italyl.buccarello@ebri.it (L.B.);; 3UOC Ophtalmology, Fondazione Policlinico Universitario A. Gemelli IRCCS, 00168 Rome, Italy; benedetto.falsini@unicatt.it; 4Department of Ophthalmology, Bambino Gesù IRCCS Children’s Hospital, 00133 Rome, Italy; 5Bio@SNS Laboratory, Scuola Normale Superiore, 56124 Pisa, Italy

**Keywords:** human NGF, mouse NGF, painless NGF, ARPE-19 cells, H_2_O_2_, UV-A, p75^NTR^ receptors, TrkA receptors

## Abstract

In spite of its variety of biological activities, the clinical exploitation of human NGF (hNGF) is currently limited to ocular pathologies. It is therefore interesting to test the effects of hNGF in preclinical models that may predict their efficacy and safety in the clinical setting of ocular disorders and compare the effects of hNGF with those of its analogs. We used a human retinal pigment cell line, ARPE-19 cells, to investigate the effects of hNGF and its analogs, mouse NGF (mNGF) and painless NGF (pNGF), on cell viability under basal conditions and after exposure to oxidative stimuli, i.e., hydrogen peroxide (H_2_O_2_) and ultraviolet (UV)-A rays. The effects of hNGF and pNGF were also tested on the gene expression and protein synthesis of the two NGF receptor subtypes, p75 neurotrophic receptors (p75^NTR^) and tyrosine kinase A (TrkA) receptors. We drew the following conclusions: (i) the exposure of ARPE-19 cells to H_2_O_2_ or UV-A causes a dose-dependent decrease in the number of viable cells; (ii) under baseline conditions, hNGF, but not pNGF, causes a concentration-dependent decrease in cell viability in the range of doses 1–100 ng/mL; (iii) hNGF, but not pNGF, significantly potentiates the toxic effects of H_2_O_2_ or of UV-A on ARPE-19 cells in the range of doses 1–100 ng/mL, while mNGF at the same doses presents an intermediate behavior; (iv) 100 ng/mL of hNGF triggers an increase in p75^NTR^ expression in H_2_O_2_-treated ARPE-19 cells, while pNGF at the same dose does not; (v) pNGF, but not hNGF (both given at 100 ng/mL), increases the total cell fluorescence intensity for TrkA receptors in H_2_O_2_-treated ARPE-19 cells. The present findings suggest a vicious positive feedback loop through which NGF-mediated upregulation of p75^NTR^ contributes to worsening the toxic effects of oxidative damage in the human retinal epithelial cell line ARPE-19. Looking at the possible clinical relevance of these findings, one can postulate that pNGF might show a better benefit/risk ratio than hNGF in the treatment of ocular disorders.

## 1. Introduction 

Nerve growth factor (NGF) was originally defined as a neurotrophic factor acting primarily on sympathetic and sensory neuronal cells [1]. At the time, it was found to exert regulatory activity in a large array of human cell types expressing NGF receptors, including skin fibroblasts, epidermal keratinocytes, umbilical vein endothelial cells and mast cells [2,3]. 

More recently, our group reported that NGF also exerts modulatory activities on mouse microglia [4], as well as on human microglial cells [5]. In spite of such a rapidly expanding profile as a pleiotropic regulatory agent, due to its potent pain-sensitizing activity [6,7], so far, hNGF has only been approved in human therapy for the topical treatment of moderate (persistent epithelial defect) or severe (corneal ulcer) neurotrophic keratitis in adults [8]. Apart from the approved indication, hNGF topical eyedrops are currently under clinical development for severe Sjogren dry eye disease [9,10], for moderate-to-severe dry eye syndrome [11,12] and for glaucoma [13].

Looking at hNGF from a pharmacological viewpoint, this agent should be considered in all respects as the first-in-class of a new class of medicines, namely the NGF analogs. Another member of this class, mNGF, shows an amino acid sequence very similar to that of hNGF (89.2%) and has been used in humans to test efficacy in skin wound healing in diabetic patients, in corneal lesions and in childhood optic gliomas [14,15,16]. Although it showed a promising benefit/risk ratio in these pilot studies (also because the lower affinity of mNGF for human TrkA and p75^NTR^ receptors might have limited the pain-sensitizing effects [17]), mNGF did not undergo further clinical development, being potentially more immunogenic than NGF as a heterologous protein, and it should only be considered as a useful pharmacological tool.

In order to overcome the liabilities of human NGF-based drug candidates and fully exploit their potential, an optimized recombinant mutated form of hNGF, the so-called pNGF, was developed and characterized [4,18]. Painless NGF harbors two amino acid changes compared to wild type hNGF: glutamic acid replacing arginine in position 100 and serine replacing proline in position 61. The former change (R100E) was inspired by the congenital painlessness disease HSAN V [19] and is aimed at reducing pain sensitization activity while preserving neurotrophic potency. The latter amino acid change (P61S) allows pNGF detection and discrimination from the endogenous NGF in biological fluids without changing its pharmacological properties. Pharmacologically, the R100E mutation reduces the binding affinity to p75^NTR^ (mainly involved in apoptosis) by two orders of magnitude compared to that of wild type NGF, with no affinity change for TrkA receptors (mediating neuronal survival [4,18]). Therefore, pNGF presents the same TrkA-mediated neurotrophic and neuroprotective properties of hNGF but shows very limited p75^NTR^ signaling, together with a markedly lower algogenic activity in in vivo animal models [20]. Painless NGF has completed a full set of preclinical studies and is currently under clinical development for vision loss in pediatric patients affected by optic nerve gliomas [21], following on the positive results of a double-blind placebo-controlled study with mNGF eyedrops [16]. 

Thus, as a common feature of the whole class of NGF analogs, their clinical exploitation as therapeutic agents focuses on the area of ocular pathologies, in spite of a vast array of biological activities. Because of such organ-specific use, it is interesting to compare the effects of hNGF and its analogs in preclinical models that may be predictive of the efficacy and safety of these agents in the clinical setting of ocular disorders. We have previously characterized a human retinal pigment cell line, ARPE-19 cells, looking in particular at their sensitivity to oxidative damage, and used this model to investigate the putative protective effects of antioxidant agents [22]. In the present study, we used ARPE-19 cells to compare the effects of hNGF, mNGF and pNGF on cell viability, both under basal conditions and after cell damage induced by exposure to standard oxidative stimuli. The expression of NGF receptors in ARPE-19 cells and their possible changes associated with oxidative damage were also investigated. 

## 2. Results

Under the experimental conditions described in Section 4, human retinal pigment ARPE-19 cells are sensitive to oxidative damage. Figure 1 shows the effects of standard oxidative agents, i.e., a 24 h exposure to 300 µM H_2_O_2_ (Figure 1B) and a 2 h exposure to UV-A (Figure 1C), compared to unchallenged control cells (Figure 1A). In this experimental paradigm, the quantification of oxidative damage, expressed as cell viability after the exposure to standardized oxidative challenge, is reported in Figure 1D, showing that the exposure to increasing doses of H_2_O_2_ for 24 h causes concentration-dependent lethality in ARPE-19. The estimated EC50 of H_2_O_2_ is between 350 and 400 µM. Likewise, the exposure to a physical oxidative stimulus, i.e., ultraviolet-A (UV-A) light, induces a time-dependent increase in ARPE-19 lethality, with an estimated EC50 achieved after an exposure to UV-A between 90 and 120 min (Figure 1E).

Under basal conditions, the exposure of ARPE-19 cells to hNGF for 24 h produces a concentration-dependent decrease in cell viability, with a significant reduction (about 24.4%) observed at 100 ng/mL (Figure 2A), whereas no effect whatsoever was observed after the exposure to pNGF up to 100 ng/mL (Figure 2B). 

Such a tendency to a cytotoxic effect associated with the exposure to hNGF, but not to pNGF, is markedly potentiated by a 24 h pre-treatment with submaximal toxic concentrations of H_2_O_2_ (Figure 3A,B). The same distinct results for hNGF and pNGF, respectively, are obtained after a 1 h pre-exposure to UV-A (Figure 3C,D). The pictures in Figure 4 are representative images showing the effects of the exposure to 100 ng/mL pNGF (Figure 4C) and 100 ng/mL hNGF (Figure 4D), both given after a 24 h pretreatment with 300 µM H_2_O_2_, compared to untreated control cells (Figure 4A) or cells pre-treated with 300 µM H_2_O_2_ only (Figure 4B). The reduced number of viable cells in the hNGF-treated cultures (Figure 4D), with respect to pNGF-incubated cells (Figure 4C), can be clearly observed. Likewise, Figure 5 shows representative images of the effects of 100 ng/mL pNGF (Figure 5C) and 100 ng/mL hNGF (Figure 5D) exposure, both given after a 1 h pre-exposure to UV-A, compared to untreated control cells (Figure 5A) or cells pre-exposed to UV-A only (Figure 5B). In this case, the distinct cytotoxic effect of hNGF versus that of pNGF (Figure 5C,D, respectively) is also clearly visible. This distinct cytotoxic effect of hNGF, with respect to pNGF, is most likely mediated by p75^NTR^ due to the greatly reduced affinity of pNGF for p75^NTR^ with respect to hNGF [4,18]. 

In a further set of experiments, we tested the effects of mNGF in the same experimental paradigm as described above. Mouse NGF appears to exert an intermediate effect compared to hNGF and pNGF, with a trend of potentiating the damage induced by H_2_O_2_ or UV-A, although to a lesser extent than hNGF (compare the result in Figure 6A,B to those in Figure 3A,C). Such ‘in-between’ behavior of mNGF is directly evident in experiments carried out with hNGF and pNGF taken as controls (Figure 6C) and is likely related to the lower binding affinity of mNGF for human p75^NTR^ than that of hNGF [17]. 

To clarify the mechanisms underlying the toxic effects of hNGF on ARPE19 cells, we investigated the expression and localization of the receptors mediating the biological actions of NGF, namely TrkA and p75^NTR^ [23,24,25,26]. In a first series of experiments carried out using RT-PCR, we found that TrkA mRNA is below the detection threshold, both in baseline conditions and after exposure to 300 µM H_2_O_2_ alone or in combination with 100 ng/mL hNGF or pNGF. In contrast, p75^NTR^ mRNA expression can be measured under baseline conditions, as well as after the exposure to 300 µM H_2_O_2_ alone. Notably, p75 mRNA expression is strongly increased after exposure to H_2_O_2_ in the presence of hNGF (Figure 7), while p75^NTR^ is not induced when H_2_O_2_ treatment is performed in the presence of pNGF (Figure 7). Given that pNGF, unlike hNGF, does not signal via p75^NTR^, this result suggests a p75-dependent positive feedback loop, whereby a p75-dependent signal increases the expression of p75^NTR^.

In order to provide complementary results to the bio-molecular experiments described above, immunofluorescence assays were performed in a separate set of experiments to assess the expression and localization of TrkA and p75^NTR^ proteins in ARPE-19 cells treated with H_2_O_2_, hNGF and pNGF. First, the immunofluorescence experiments showed that ARPE-19 cells express TrkA receptors (visualized with the mouse monoclonal antibody MNAC13 directed against the native human receptor) and p75^NTR^ receptors (Figure 8B,C). The negative controls omitting the primary antibodies are shown in Appendix A. With regard to the subcellular localization of the two receptors, in physiological/control conditions, p75^NTR^ was detected mostly in the perinuclear zone and the nuclear zone (Figure 8B), with the latter associated with the presence of p75+ clusters of puncta in the nuclear portion (red dots, Figure 8B), whereas we observed a cytosolic or nuclear localization for TrkA receptors associated with small dots in the cytoplasmic compartment (small green dots, Figure 8C). 

In cells treated with H_2_O_2_ (Figure 8E–H), the nuclear p75^NTR^ clusters of puncta disappeared and were replaced by a more diffuse localization in the nucleus, with a stronger accumulation in the perinuclear membrane (Figure 8F). With regard to TrkA receptor labeling, following oxidative stress induction, the TrkA receptor labeling appeared more intense than that of p75^NTR^, compared to the control condition (Figure 8A–H). This observation was confirmed through the quantitative analysis of immunofluorescence (Figure 9A,B). Besides being more intense, the subcellular localization of TrkA receptors after H_2_O_2_ treatment was also mostly nuclear and cytosolic, with a greater increase in the perinuclear zone (green dots, Figure 8G).

The distinct effects of hNGF and pNGF on H_2_O_2_-treated ARPE-19 cells were then compared. The addition of hNGF to H202-treated cells significantly increased the p75^NTR^ expression compared to H_2_O_2_ exposure alone (*p* < 0.001, Figure 8F,J and Figure 9B), while pNGF-H_2_O_2_ treated cells showed a strong reduction in the p75 labeling (*p* < 0.001, Figure 8N and Figure 9B). Conversely, in H_2_O_2_-treated ARPE-19 cells incubated with pNGF, the intensity of TrkA receptor expression was higher than that in the cells incubated with hNGF (*p* < 0.001 vs. *p* < 0.01; Figure 8K,O and Figure 9A). Moreover, the p75/TrkA co-localization was reduced by pNGF after H_2_O_2_ treatment (Figure 8D vs. Figure 8H).

We conclude that pNGF induces a marked decrease in p75^NTR^ expression in H_2_O_2_-treated ARPE-19 cells, whereas in the same cells, hNGF increases p75^NTR^ expression. These data corroborate the above RT-PCR results, pointing to an increase in both p75^NTR^ mRNA and protein driven by hNGF treatment (Figure 7).

## 3. Discussion

The main findings of this study are the following: (i) the human retinal pigment cell line ARPE-19 is sensitive to oxidative damage since exposure to H_2_O_2_ or UV-A causes a concentration- or time-dependent decrease in the number of viable cells in vitro; (ii) under baseline control conditions, hNGF, but not its mutant optimized form pNGF, causes a concentration-dependent decrease in cell viability in the 1–100 ng/mL range; (iii) hNGF significantly potentiates the toxic effects of H_2_O_2_ or UV-A on ARPE-19 cells, whereas pNGF has no influence whatsoever on the damaging effects of oxidative agents, while mNGF presents an intermediate behavior in this experimental setting; (iv) hNGF triggers an increase in p75^NTR^ expression in H_2_O_2_-treated ARPE-19 cells, while pNGF does not; (v) pNGF, more so than hNGF, increases the total cell fluorescence intensity for TrkA receptors in H_2_O_2_-treated ARPE-19 cells.

Oxidative damage to the retinal pigment epithelium (RPE) is highly relevant to inherited retinal degenerations (IRDs) and age-related macular degeneration (AMD). The RPE is a crucial cellular layer providing essential support and maintenance functions for the photoreceptors. It plays a vital role in the visual cycle, nutrient transport, waste disposal and protection against oxidative stress [27]. Diseases such as retinitis pigmentosa, Stargardt disease and AMD often involve dysfunction or loss of RPE cells [28,29]. Oxidative damage significantly contributes to the degenerative process in these diseases. Several factors may underlie oxidative stress in the RPE, such as light exposure, retinal metabolism and impaired antioxidant defense mechanisms. Oxidative damage to the RPE can lead to various detrimental effects, including lipofuscin accumulation, mitochondrial dysfunction, impaired phagocytosis of photoreceptor outer segments and inflammation. These processes contribute to the progressive degeneration of photoreceptors and vision loss [27,28,29].

Given the broad neuroprotective properties of NGF and NGF-related proteins, exerted on target neurons and on glial cells, and the availability of clinically approved hNGF eyedrops [30], there is interest in evaluating both the neuroprotective effects of NGF and also its potential safety pitfalls and liabilities in human retinal cell models. In turn, this may help with designing and exploiting optimized NGF-related molecules.

In the present study, we found that hNGF worsened the H_2_O_2_-mediated (as well as the UV-mediated) oxidative damage of human RPE cells in a dose-dependent manner. We surmise that this exacerbation may be linked to the enhanced expression of the proapoptotic receptor p75^NTR^ induced by hNGF, for the following reasons: (i) pNGF, which is a TrkA-biased variant of NGF with a 200-fold lower binding affinity for p75^NTR^, does not exacerbate the oxidative damage effect; (ii) hNGF, but not the p75-less pNGF, is a potent inducer of p75^NTR^ mRNA and protein in the human retinal ARPE-19 cell line. Based on the data presented, this suggestion is in line with the established fact that the expression of p75^NTR^ is upregulated by different forms of injuries and insults in different cells and tissues [31,32]. Moreover, oxidative stress induces p75^NTR^-mediated neurite neurodegeneration and apoptosis [33]. Notably pNGF, a TrkA-biased mutant NGF agonist devoid of p75^NTR^ signaling ability, affected neither the H_2_O_2_-treated RPE cell viability nor the p75^NTR^ expression level. On the other hand, as far as neuroprotection against oxidative stress is concerned, we observed that pNGF, while showing no additional toxic effect, did not ameliorate the reduced cell viability, as might have been expected. This lack of neuroprotection could arise due to the low expression of TrkA receptors in these cells. Compelling evidence showed that NGF and its mutated form exert neuroprotection by acting on glial cells [4,18]. Hence, a diverse cellular landscape, in its full complexity as found in vivo, is essential in order to obtain effective non-cell-autonomous neuroprotection of human retinal epithelial target cell(s) by pNGF. Future experiments on the effects of pNGF on H_2_O_2_-treated ARPE-19 cells co-cultured with different types of retinal glial cells will allow testing whether neuroprotection by pNGF, against oxidative stress toxicity of human epithelial cells, can be observed in these conditions. 

Both NGF and its precursor proNGF are expressed in the rat RPE [34] and NGF mRNA, and p75^NTR^ expression was also demonstrated in both human iris epithelial cells and RPE cells [35]. While the expression of the NGF and its p75^NTR^ receptor in RPE cells has been previously observed, little is known about the functional significance of NGF-p75^NTR^ signaling in RPE physiology and its potential role in retinal diseases. Our results provide a significant functional contribution to the understanding of NGF-p75 receptor signaling in a human retinal epithelial cell line. Some studies suggested that NGF may exert direct neuroprotective action on retinal photoreceptors [36,37]. A recent study in a zebrafish model of retinal degeneration [38] showed that intravitreally injected NGF has a pro-regenerative effect on photoreceptors. Depending on the hNGF dose, all these effects may be dampened by the p75-mediated pro-apoptotic effect. The use of pNGF, without influence on RPE cell death, could exert full efficacy in its neuroprotection of photoreceptors. The use of pNGF could potentiate the effect of antioxidants currently under evaluation to treat cone degeneration in IRD, i.e., n-acetylcysteine [39], which has been shown to protect RPE cells from oxidative damage [40]. 

It is important to take into account the limitations of this study. We consider that all data presented here have intrinsic validity, since different analogs are compared with each other in a system showing consistent responses to standardized stimuli. However, the system is limited to a single cell line, namely ARPE-19 cells. These cells have been extensively used since their first description in 1996, and more than 2000 reports have been published so far based on this cell line. A recent review has been published that analyzed the pitfalls of ARPE-19 cells used as a model of retinal pigmented epithelium [41], suggesting that a confirmation of the present results in a model of primary, non-transformed RPE cells is advisable. 

## 4. Materials and Methods

### 4.1. Materials

Human recombinant NGF was purchased from the Alomone laboratory (Jerusalem BioPark, Jerusalem, Israel). Painless NGF was kindly provided by Chiesi Farmaceutici S.p.A. (Parma, Italy). Hydrogen peroxide solution 30% (*w*/*w*) in H2O, 100 mL vials, and mouse NGF were purchased from Merck Life Science S.r.L. (Milan, Italy). 

### 4.2. Cell Cultures and Treatments

ARPE-19 cells were obtained from the American Type Cell Culture (ATCC-CRL-2302, Manassas, VA, USA) and cultured according to the manufacturer’s instructions. The cells were raised in a DMEM/F12 medium (Sigma-Aldrich, St. Louis, MO, USA) supplemented with 10% FBS (Gibco; Thermo Fisher Scientific Inc., Waltham, MA, USA), 2mM L-Glutamine (Sigma-Aldrich, St. Louis, MO, USA) and 100 U/mL penicillin–streptomycin (Thermo Fisher Scientific Inc., Waltham, MA, USA) at 37 °C in a 5% CO_2_ environment. When cells reached 80% of confluence, they were split and sub-cultured at a concentration of 30,000 cells/cm^2^ at first, and later at a density of 15,000 cells/well in 96-well plates for experimental procedures.

On the day of the experiment, the cells were pre-treated for 24 h, in starvation conditions (0% FBS), with hNGF, pNGF or mNGF (treated groups) or medium alone (control group). The pre-treatment was followed by a second incubation period of 24 h, during which cells were exposed to oxidative stress in the presence of the neurotrophic factor added again.

The cells were exposed to two different agents inducing oxidative stress: H_2_O_2_ and UV-A rays, a chemical and physical oxidative stimulus, respectively. Specifically, after pretreatment with the neurotrophic factor (hNGF, pNGF or mNGF), the cells were exposed to H_2_O_2_ for 24 h or UV-A rays for 1 h. Irradiation was performed using a UV lamp (Vilber Lourmat VL-62C Power 6W; Vilber Lourmat Deutschland GmbH, Eberhardzell, Germany) with wavelength at 365 nm, placed at 10 cm from the cells for 1 h at an intensity of approximately 0.06 J/cm^2^/s. Immediately after exposure to the UV-A rays, the cells were put in an incubator until the end of the 24 h incubation period.

### 4.3. Assessment of Cell Viability

Twenty-four hours after inducing oxidative stress, with or without the neurotrophic factor, the cell viability was evaluated using the MTS assay (Promega, Waltham, MA, USA), according to the manufacturer’s procedure. The MTS reagent (20 µL) was added to cells, where it was converted to formazan. The quantity of formazan released into the culture supernatant, directly proportional to the number of living cells, was determined by measuring the absorbance at 490 nm with a microplate photometer (Victor 4, PerkinElmer, Waltham, MA, USA). The results are expressed as the percentage of cell viability relative to the untreated control. 

Furthermore, the morphological features of the ARPE-19 cells, exposed to oxidative damage and/or in the presence of neurotrophic factors, were analyzed and photographed using phase-contrast microscopy (TE300-Eclipse-microscope; Nikon Corporation, Tokyo, Japan) at a magnification of ×10.

### 4.4. Real-Time Quantitative RT-PCR (qRT-PCR) Analysis

Total RNA from ARPE-19 was extracted using the Trizol reagent protocol. A Qubit™ RNA HS Assay Kit was used to measure RNA concentration (Thermo Fisher Scientific), and 40 ng aliquots of RNA were converted to cDNA using random hexamer primers. Quantitative changes in mRNA levels were measured through qRT-PCR using the following conditions: 35 cycles of denaturation (95 °C for 20 s), annealing and extension (60 °C for 20 s). The qRT PCR was carried out using the Brilliant III Ultra-Fast SYBR^®^ Green QPCR Master Mix (Stratagene, San Diego, CA, USA). PCR reactions were conducted in a 20 µL reaction volume using an AriaMX real-time PCR machine (Agilent, Santa Clara, CA, USA). The following primer sequences were used: P75^NTR^ forward primers CCTACGGCTACTACCAGGATG, reverse primers CACACGGTGTTCTGCTTGT; human TrkA receptor forward primers TCAATGGCTCCGTGCTCAAT, reverse primers TGCTGTTAGTGTCAGGGATGG and β-actin forward primers ACGTTGCTATCCAGGCTGTGCTAT, reverse primers TTAATGTCACGCACGATTTCCCGC. Relative mRNA concentrations were calculated from the take-off point of reactions (threshold cycle, Ct comparative quantitation) using AriaMX software (Agilent Aria v1.5) and based upon the −∆∆Ct method. Ct values for β-actin expression worked as a normalizing signal.

### 4.5. Immunofluorescence

Cells were plated on ibidi chambers at a density of 45,000 cells/cm^2^. After the treatment, the seeding medium was removed, and cells were fixed in 4% paraformaldehyde-PBS for 20 min at room temperature. After three washes in PBS 1X (10 min each), cells were permeabilized in PBS-Triton X-100 (Fluka) 0.5% for 10 min at room temperature and then washed three times with PBS 1X (10 min each). Cells were incubated for 1 h at room temperature with blocking solutions containing 0.3% Triton X-100 plus 2% bovine serum albumin (BSA) and 0.1 M Glycine, then overnight at 4 °C with the following primary antibodies: polyclonal rabbit anti-p75 (1:500, Promega cod. G3231) and mouse monoclonal anti-TrkA, MNAC13 (1:500 from 93 γ/λ stock concentration). The fluorescence was detected using secondary antibodies conjugated with Alexa-Fluor 488 (Green) and Alexa-Fluor 546 (Red) (1:500; Invitrogen-Thermo Fisher, Segrate, MI, Italy, cod. A32723 and A31572) in PBS for 1 h at RT. Each well was then incubated in Dapi (1:500; Invitrogen-Thermo Fisher) in PBS to make the cell nuclei visible. Coverslips were mounted using Fluoromount mounting medium (Sigma, cod. F4680). Confocal images were acquired with an Olympus microscope equipped with an Olympus Confocal scan unit (microscope BX61 and Confocal system FV500) managed using AnalySIS Fluoview software with 3 laser lines, a UV diode laser (405 nm), Ar–Kr (488 nm) and He–Ne (546 nm), respectively, used to detect Dapi staining and secondary antibodies. Double staining was revealed with a scanning sequential mode to eliminate possible bleed-through effect. For the negative control, only secondary antibodies were incubated in PBS for 1 h at RT. Quantitative data from images were obtained keeping the following image acquisition criteria: 40× objective, 1024 × 1024 frame.

The intensity of p75^NTR^ and MNAC13 immunofluorescence was quantified following the protocol described by L. Hammond, QBI, The University of Queensland, Australia. The corrected total cell fluorescence (CTCF) was calculated by subtracting the value corresponding to the integrated density from the cell’s area for the mean fluorescence of background readings. A total of 10 cells were counted in each field, and 5 fields were examined for each experimental condition (control, H_2_O_2_, hNGF and pNGF), yielding a total of 50 cells analyzed for each group (n = 50).

### 4.6. Statistical Analysis

Each experiment was repeated a minimum of two times, in sextuplicate. Data were analyzed through one-way analysis of variance (ANOVA), followed by the post hoc Dunnet or Newman–Keuls tests for comparisons between group means. Immunofluorescence quantification was analyzed using one-way ANOVA, followed by Bonferroni’s post hoc test. Data were analyzed using a Prism^TM^ computer program (GraphPad, San Diego, CA, USA). All data are expressed as means ± 1 standard error of the mean (SEM), and differences were considered statistically significant if *p* < 0.05.

## 5. Conclusions

NGF and its analogs display a vast array of biological activities; nevertheless, the clinical exploitation of hNGF is currently limited to the setting of ocular disorders. In this study, we showed that hNGF, but not the optimized variant ‘painless NGF’, exacerbates oxidative damage in human epithelial retinal pigment cells; hNGF toxicity appears to be associated with a vicious cycle of NGF-induced over-expression of p75^NTR^ receptors. We conclude that painless NGF might show a better benefit/risk ratio than NGF in the treatment of ocular disorders.

## Figures and Tables

**Figure 1 ijms-24-16237-f001:**
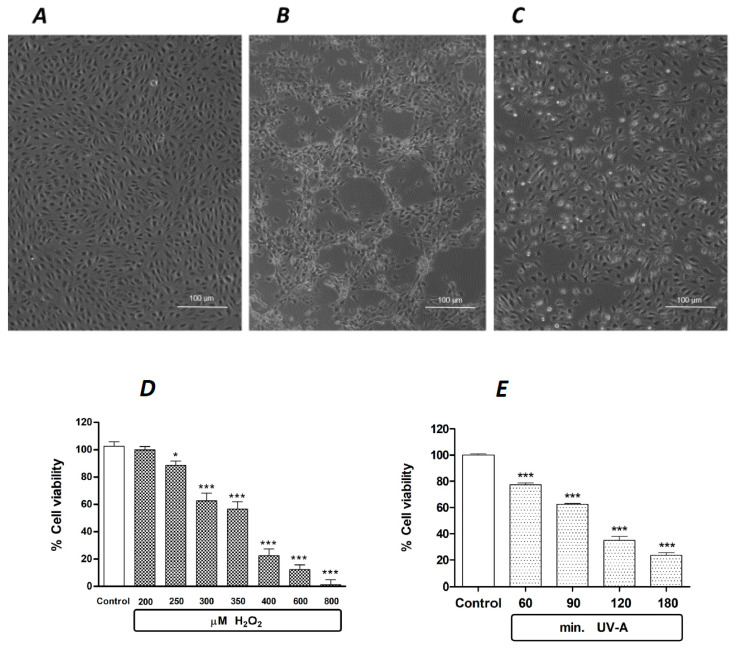
The effects of standard oxidative stimuli, H_2_O_2_ and UV-A on human ARPE-19 cells in vitro. Representative phase contrast microscopies (×10 magnification) of human ARPE-19 cells under baseline control conditions (**A**), after a 24 h exposure to 300 µM H_2_O_2_ (**B**) and after a 2 h exposure to UV-A (**C**). (**D**): H_2_O_2_ decreases ARPE-19 cell viability in a concentration-dependent manner after a 24 h exposure. (**E**): UV-A decreases ARPE-19 cell viability in a time-dependent manner. Data from (**D**,**E**) are expressed as percentage cell viability, with the means ± 1 SEM of 6 replicates per experimental group. *: *p* < 0.05 and ***: *p* < 0.001 vs. controls.

**Figure 2 ijms-24-16237-f002:**
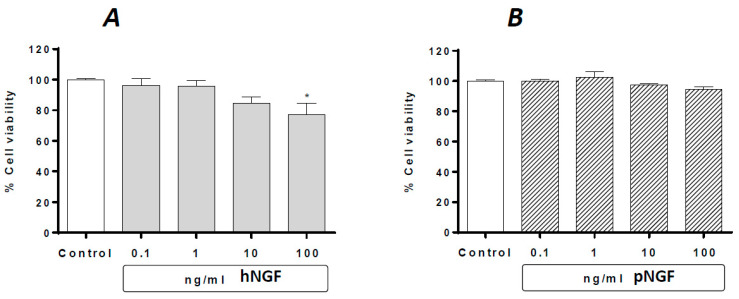
The effects of hNGF (**A**) and pNGF (**B**) on cultured human ARPE-19 under baseline conditions. Data are expressed as percentage cell viability, with the means ± 1 SEM of 6 replicates per experimental group. *: *p* < 0.05 vs. controls.

**Figure 3 ijms-24-16237-f003:**
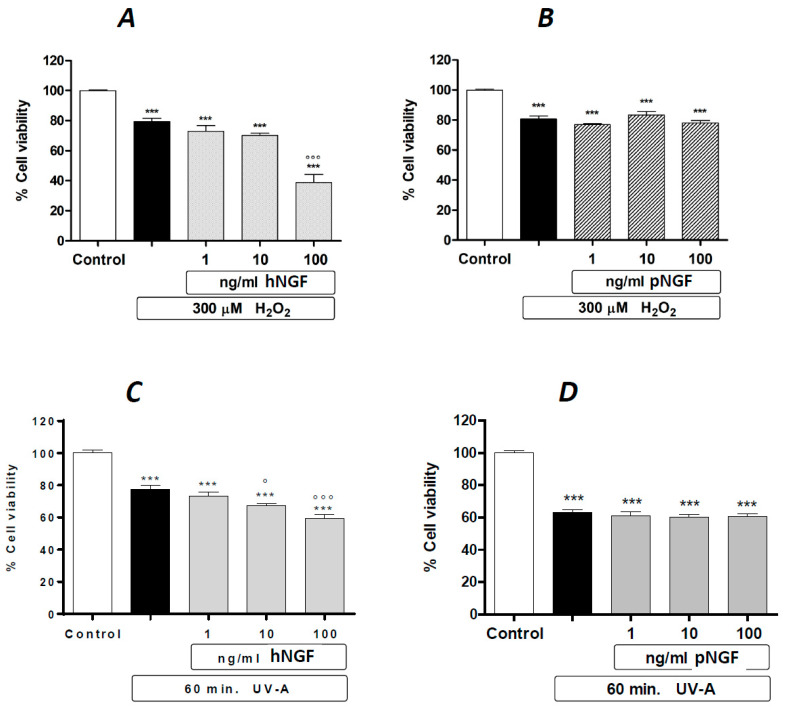
The effects of hNGF and pNGF on cultured human ARPE-19 after a 24 h exposure to 300 µM H_2_O_2_ (**A**,**B**) or a 2 h exposure to UV-A (**C**,**D**). Black bars: H_2_O_2_ (**A**,**B**) or UV-A (**C**,**D**) given alone. Data are expressed as percentage cell viability, with the means ± 1 SEM of 6 replicates per experimental group. ***: *p* < 0.001 vs. controls. ° and °°°: *p* < 0.05 and *p* < 0.001 vs. the oxidative stimulus given alone, respectively.

**Figure 4 ijms-24-16237-f004:**
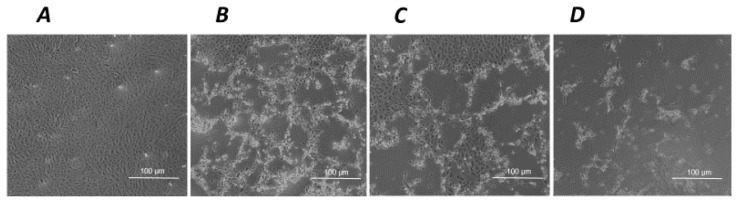
The effects of hNGF and pNGF on cultured human ARPE-19 after a 24 h exposure to 300 µM H_2_O_2_. Representative phase contrast microscopies (×10 magnification) showing human ARPE-19 cells under baseline control conditions (**A**) and after a 24 h exposure to 300 µM H_2_O_2_ alone (**B**) or in the presence of 100 ng/mL pNGF (**C**) or 100 ng/mL hNGF (**D**).

**Figure 5 ijms-24-16237-f005:**
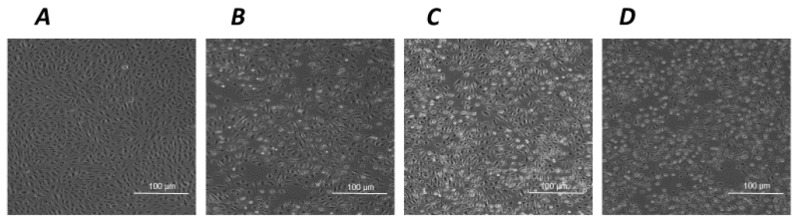
The effects of hNGF and pNGF on cultured human ARPE-19 after a 2 h exposure to UV-A. Representative phase contrast microscopies (×10 magnification) showing human ARPE-19 cells under baseline control conditions (**A**) and after a 2 h exposure to UV-A alone (**B**) or in the presence of 100 ng/mL pNGF (**C**) or 100 ng/mL hNGF (**D**).

**Figure 6 ijms-24-16237-f006:**
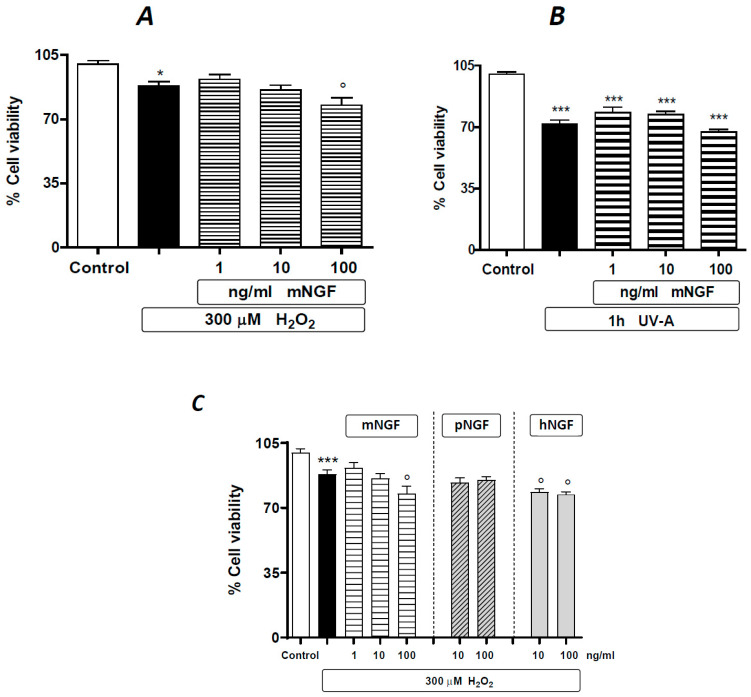
The effects of mNGF on human ARPE-19 cells under baseline conditions and after oxidative stimuli. The effects of graded doses of mNGF on ARPE-19 cells pre-exposed to 300 µM H_2_O_2_ (**A**) or to UV-A (**B**). (**C**) Comparative effects of mNGF, pNGP and hNGF on ARPE-19 cells pre-exposed to 300 µM H_2_O_2_. Black bars: H_2_O_2_ (**A**,**C**) or UV-A (**B**) given alone. Data are expressed as percentage cell viability, with the means ± 1 SEM of 6 replicates per experimental group. * and ***: *p* < 0.05 and *p* < 0.001 vs. controls, respectively. °: *p* < 0.05 vs. the oxidative stimulus given alone.

**Figure 7 ijms-24-16237-f007:**
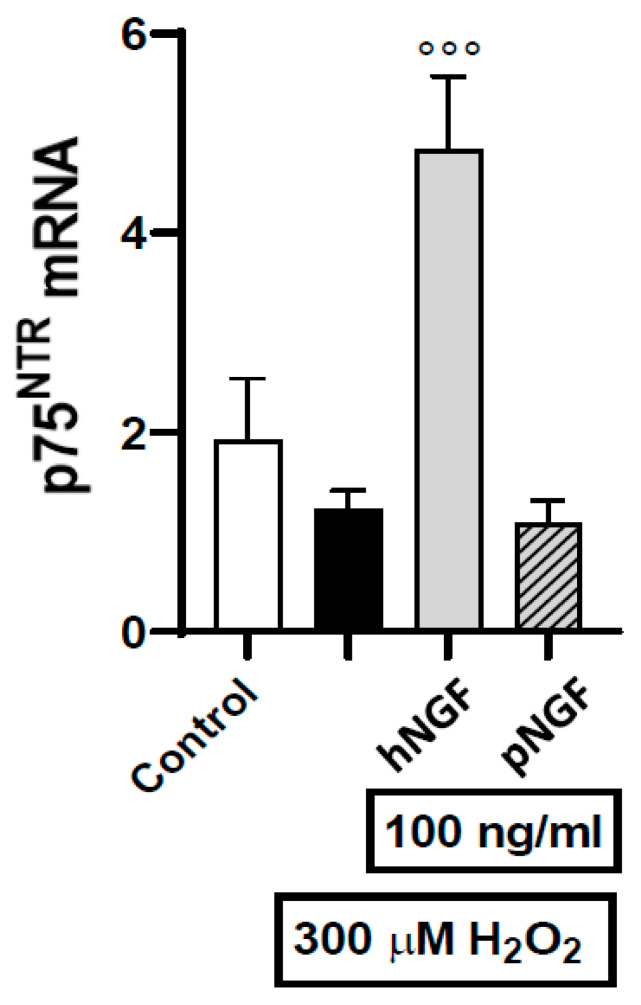
The effects of 100 ng/mL hNGF and pNGF on p75^NTR^ gene expression in ARPE-19 cells pre-exposed to 300 µM H_2_O_2_. Black bar: H_2_O_2_ given alone. Data are the means ± 1 SEM of the results from 3 different experiments with similar results. °°°: *p* < 0.001 vs. the oxidative stimulus given alone.

**Figure 8 ijms-24-16237-f008:**
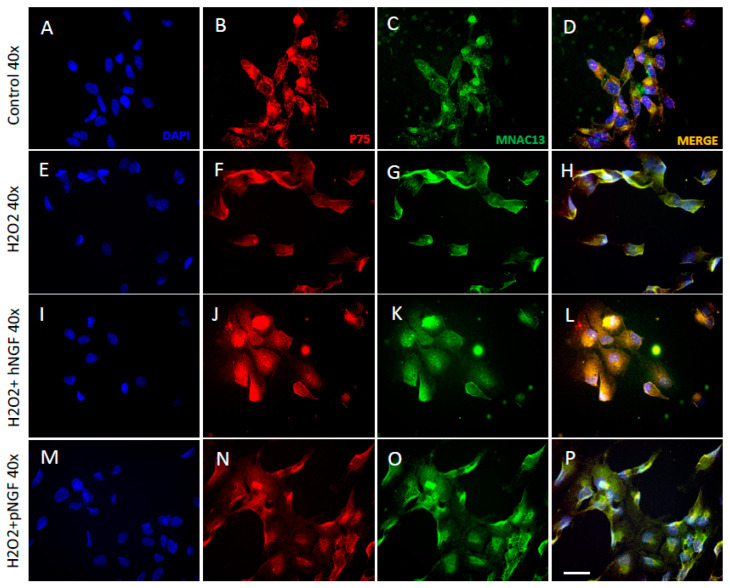
Expression of NGF receptors in human ARPE-19 cells under baseline conditions (**A**–**D**) and after exposure to 300 µM H_2_O_2_ alone (**E**–**H**) or in the presence of hNGF (**I**–**L**) or pNGF (**M**–**P**). Blue fluorescence indicates DAPI staining of cell nuclei, red and green fluorescence indicate p75^NRT^ and TrkA expression, respectively. Magnification ×40.

**Figure 9 ijms-24-16237-f009:**
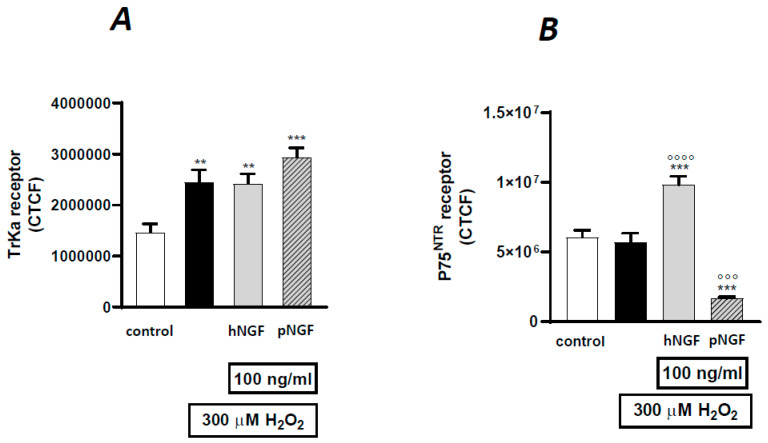
The correct total cell fluorescence (CTCF) of TrkA (**A**) and p75^NRT^ (**B**) receptor expression in ARPE-19 cells under baseline conditions or after exposure to 300 µM H_2_O_2_, alone or in the presence of 100 mg/mL hNGF or pNGF. Black bars: H_2_O_2_ given alone. Data are the means ± 1 SEM of the results from 3 different experiments with similar results. ** and ***: *p* < 0.01 and *p* < 0.001 vs. controls; °°° and °°°°: *p* < 0.001 and *p* < 0.0001 vs. the oxidative stimulus given alone.

## Data Availability

Data are available on request from the authors.

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
