# Peer review of "A Vicious NGF-p75NTR Positive Feedback Loop Exacerbates the Toxic Effects of Oxidative Damage in the Human Retinal Epithelial Cell Line ARPE-19"

_ijms, 2023, doi:10.3390/ijms242216237_

Round 1

Reviewer 1 Report

Comments and Suggestions for Authors

Manuscript Overview:  Tringali G et al. conducted experiments on human retinal epithelial ARPE-19 cells to investigate the effects of nerve growth factor (NGF) and its analogs (mouse NGF and painless NGF) on cell viability under oxidative stress induced by hydrogen peroxide (H2O2) and ultraviolet (UV)-A rays, compared to cells exposed to basal conditions.  The authors report that oxidative stress resulted in reduced viability, as did treatment with NGF, but not painless NGF.  NGF also triggered increased p75NTR.  In contrast, painless NGF but not hNGF increased TrKA in H2O2-exposed cells.  The authors concluded that hNGF, but not painless hNGF exacerbates oxidative stress in ARPE-19 cells via p75NTR.

 Comments to the authors:

Abstract: 

1.      State the working hypothesis.

2.      Spell out abbreviations at first mention and maintain consistency throughout (NGF, H2O2, UV, NTR).

3.      There is confusion regarding NGF.  In the methods, the authors state that NGF and its analogs, mouse NGF and painless NGF were used.  However, in the conclusions, the authors refer to painless hNGF and hNGF.  Clarify and maintain consistency, when using acronyms.  Acronyms should be spelled out at first mention to avoid confusion.

4.      In the title, NTR is not superscript, but it is in the Abstract.  Maintain consistency.

5.      The conclusions is a rehashing of the findings.  The conclusions should clearly state the meaning of the findings.  What is the clinical significance?

Introduction: 

1.      The sentence structure on lines 38-39 needs revising.

2.      The authors refer to painless NGF as pNGF, not consistent with the Abstract.

3.      Clearly state the working hypothesis.

4.      Similar conclusions on page 3, lines 93-98 should be stated in the Abstract.

Results:

1.      The authors should provide further clarity in the figure legends regarding the different bars. In particular, there is no label for the black bars for figures 3, 6, 7, 9. 

Discussion:

2.      The authors state that hNGF significantly potentiates the toxic effects of H2O2 or UV-A, but pNGF has not effects.  While this is true for cells exposed to basal conditions, Figure 3 clearly shows that oxidative stress induced cells treated with H2O2, and UV-A had significantly decreased cell viability in a similar manner to hNGF.  This was also seen in the mNGF groups exposed to UV-A (Figure 6B).

3.      The authors also stated that only pNGF and not hNGF increased TrkA fluorescence in H2O2-exposed cells.  However, Figure 9 shows that TrkA receptor was induced in all groups compared to controls, but it was the p75NTR receptor that was higher with only hNGF.

4.      Maintain consistency for TrkA or TrKa receptor; as well as p75NTR or p75NTR receptor. 

5.      It is important to state the limitations of the study. 

Materials and Methods: 

1.      Provide a justification and clinical relevance for the doses of hNGF, mNGF, and pNGF.

Comments on the Quality of English Language

Minor corrections.

Author Response

REPLIES TO REVIEWER 1

Abstract:

  1. State the working hypothesis.

The working hypothesis has been re-written more clearly, in order to stress the concept that a comparison has been carried out between hNGF and its analogs, with emphasis on painless NGF.

  1. Spell out abbreviations at first mention and maintain consistency throughout (NGF, H2O2, UV, NTR).

Abbreviations throughout the abstract have been amended according to referee’s suggestions.

  1. There is confusion regarding NGF. In the methods, the authors state that NGF and its analogs, mouse NGF and painless NGF were used. However, in the conclusions, the authors refer to painless hNGF and hNGF. Clarify and maintain consistency, when using acronyms. Acronyms should be spelled out at first mention to avoid confusion.

Misspelling in the conclusion has been amended. We wish to clarify that mouse NGF has been used as a pharmacological tool but cannot be used in humans, therefore it has no clinical significance and has not been mentioned in the conclusions.

  1. In the title, NTR is not superscript, but it is in the Abstract. Maintain consistency.

NTR is now superscript in the title as well.

  1. The conclusions is a rehashing of the findings. The conclusions should clearly state the meaning of the findings. What is the clinical significance?

The conclusions have been completed with a statement on the clinical significance of the findings presented in the paper.

Introduction:

  1. The sentence structure on lines 38-39 needs revising.

The sentence has been revised following on the referee’s request.

  1. The authors refer to painless NGF as pNGF, not consistent with the Abstract.

Previous inconsistencies between introduction and abstract have been amended.

  1. Clearly state the working hypothesis.

The working hypothesis of the study was the comparison between human NGF and its analogs (with special emphasis on painless NGF) on a cellular model of pigmented epithelium. The gene expression and synthesis of MGF receptors in these cells was also investigated to explain the results of functional experiments. This hypothesis is clearly stated in the introduction (last 4 lines from bottom, in the revised version).

  1. Similar conclusions on page 3, lines 93-98 should be stated in the Abstract.

The conclusions at lines 93-98 have been included in the Abstract. However, please note that lines 90-98 of previous version have been removed to comply with the suggestion of Referee n. 2.

Results:

  1. The authors should provide further clarity in the figure legends regarding the different bars. In particular, there is no label for the black bars for figures 3, 6, 7, 9.

Further descriptions concerning the black bars have been provided (see Figure Legends)

Discussion:

  • The authors state that hNGF significantly potentiates the toxic effects of H2O2 or UV-A, but pNGF has not effects. While this is true for cells exposed to basal conditions, Figure 3 clearly shows that oxidative stress induced cells treated with H2O2, and UV-A had significantly decreased cell viability in a similar manner to hNGF. This was also seen in the mNGF groups exposed to UV-A (Figure 6B).

Figure 3 shows that both oxidative stimuli (black bars) significantly reduce cell viability (p<0.001; 3 asterisks) compared to unchallenged control cells (with bars). Figure 3 also shows that the addition of hNGF to the oxidative stimuli further reduces viability at 10 ng/ml (p<0.05, 1 open circle – panel C) and 100 ng/ml (p<0.001, 3 open circles - panels A and C). The same does not occur with pNGF given in addition to the oxidative stimuli, and indeed no open circles are drawn. Although at first glance it might appear that no differences exist between hNGF and pNGF, the statistical analysis clearly says the contrary. The same applies to results shown in figure 6.

  • The authors also stated that only pNGF and not hNGF increased TrkA fluorescence in H2O2-exposed cells. However, Figure 9 shows that TrkA receptor was induced in all groups compared to controls, but it was the p75NTR receptor that was higher with only hNGF.

The referee is right: TkrA expression was induced in all groups compared to untreated controls. However, where we state that only pNGF and not hNGF increase TrkA fluorescence, we are referring to controls exposed to H2O2 (black bars) and not to unexposed controls (with bars). Thus, there is no contradiction between our statement and the results shown in figure 9A.

  • Maintain consistency for TrkA or TrKa receptor; as well as p75NTR or p75NTR receptor.

Consistency between TrkA and TrkA receptor; as well as between p75NTR and p75NTR receptor has been checked throughout the text.

  • It is important to state the limitations of the study.

A new paragraph has been added discussing the limitations of the study (last 8 lines from bottom).

Materials and Methods:

  1. Provide a justification and clinical relevance for the doses of hNGF, mNGF, and pNGF.

By selecting the concentrations of NGF and its analogs to be used in the study (namely 1-100 ng/ml), we take into account mostly the range of concentrations used for these agents in comparable in vitro paradigms, i.e., experiments on cell cultures. We did not consider the information concerning NGF plasma levels measured in vivo in man (which are in the 10-100 pg/ml range) since this approach might be misleading. In fact, circulating levels under physiological conditions might not reflect the levels achieved in ocular tissues after topic administration.

Reviewer 2 Report

Comments and Suggestions for Authors

Dear authors,

Overall, the manuscript is complex and contain new valuable and interesting information’s. The title of the manuscript accurately describes its subject. The experiments are well planned and executed with encouraging results, the literature cited is up to date. I find the topic of this study is worthwhile and of great interest .

There are some negative aspects, which I believe need to be clarified and corrected in order to publish this manuscript. 

Abstract

The abstract should be self-explanatory. Authors should mention doses used.

Introduction

Lines 90 – 98 – This paragraph, in my opinion, doesn't belong in this chapter. It appears to be a conclusion as it is presented.

Results

Figure 1A, B, C – All staining images are sub-standard. It is suggested to enhance those and include high resolution images. The same observation applies to Figures 4 and 5, respectively.

Discussions

The discussions are presented in a proper manner, with reference to recent literature data.

To enhance the overall quality of the manuscript and high visibility of the reported results, it is suggested to include a schematic diagram in the discussion section delineating the mechanistic overview of  the effects of hNGF and pNGF on H2O2-treated ARPE-19 cells.

Materials and Methods

When it comes to the methodology, I really liked how well-organized everything was and how many tests were performed.

Lines 329 – 333  – Provide information about concentrations of hNGF, pNGF, or else mNGF (treated groups). When conducting research on cell cultures, more specific information about the quantities of other chemicals and specific concentrations needs to be provided.

Author Response

REPLIES TO REVIEWER 2

Abstract

The abstract should be self-explanatory. Authors should mention doses used.

The doses used where added into the abstract where appropriate

Introduction

Lines 90 – 98 – This paragraph, in my opinion, doesn't belong in this chapter. It appears to be a conclusion as it is presented.

Lines 90-98 of the introduction were removed, as requested by the referee.

Results

Figure 1A, B, C – All staining images are sub-standard. It is suggested to enhance those and include high resolution images. The same observation applies to Figures 4 and 5, respectively.

Figures 1, 4 and 5 have now a 300-DPI high resolution. However, please note that these are phase-contrast images, and no staining technique was applied.

Discussions

The discussions are presented in a proper manner, with reference to recent literature data.

To enhance the overall quality of the manuscript and high visibility of the reported results, it is suggested to include a schematic diagram in the discussion section delineating the mechanistic overview of the effects of hNGF and pNGF on H2O2-treated ARPE-19 cells.

A scheme has been prepared according to referee suggestion. The scheme has been uploaded as supplementary material; we would like to leave to the editorial Office the decision if and where to insert the scheme within the main text.

Materials and Methods

When it comes to the methodology, I really liked how well-organized everything was and how many tests were performed.

Lines 329 – 333 – Provide information about concentrations of hNGF, pNGF, or else mNGF (treated groups). When conducting research on cell cultures, more specific information about the quantities of other chemicals and specific concentrations needs to be provided.

See reply to Referee 1 on the same question.
